# A fully protected hydrogenase/polymer-based bioanode for high-performance hydrogen/glucose biofuel cells

Adrian Ruff [1], Julian Szczesny[1], Nikola Marković[1], Felipe Conzuelo [1], Sónia Zacarias[2], Inês A.C. Pereira [2], Wolfgang Lubitz[3] & Wolfgang Schuhmann [1]

Hydrogenases with Ni- and/or Fe-based active sites are highly active hydrogen oxidation catalysts with activities similar to those of noble metal catalysts. However, the activity is connected to a sensitivity towards high-potential deactivation and oxygen damage. Here we report a fully protected polymer multilayer/hydrogenase-based bioanode in which the sensitive hydrogen oxidation catalyst is protected from high-potential deactivation and from oxygen damage by using a polymer multilayer architecture. The active catalyst is embedded in a low-potential polymer (protection from high-potential deactivation) and covered with a polymer-supported bienzymatic oxygen removal system. In contrast to previously reported polymer-based protection systems, the proposed strategy fully decouples the hydrogenase reaction form the protection process. Incorporation of the bioanode into a hydrogen/glucose biofuel cell provides a benchmark open circuit voltage of 1.15 V and power densities of up to 530 μW cm$^{-2}$ at 0.85 V.

---

[1] Analytical Chemistry - Center for Electrochemical Sciences (CES), Ruhr-Universität Bochum, Universitätsstr. 150, Bochum D-44780, Germany. [2] Instituto de Tecnologia Química e Biológica António Xavier, Universidade Nova de Lisboa, Oeiras 2780-157, Portugal. [3] Max-Planck-Institut für Chemische Energiekonversion, Stiftstrasse 34–36, Mülheim an der Ruhr 45470, Germany. Correspondence and requests for materials should be addressed to A.R. (email: adrian.ruff@ruhr-uni-bochum.de) or to W.S. (email: wolfgang.schuhmann@ruhr-uni-bochum.de)

Hydrogenases are nature´s highly efficient biocatalysts for the reversible conversion of $H_2$ into protons[1]. Their rate constants for $H_2$ oxidation are similar to those of scarce and costly noble-metal-based catalysts like Pt[2,3]. In the late 1970s they were proposed as promising alternatives for the fabrication of high current-density ($J$) $H_2$-oxidizing anodes either in a direct electron transfer (DET) or a mediated electron transfer (MET) regime[4]. In combination with an $O_2$-reducing biocatalyst, typically the multicopper enzymes bilirubin oxidase or laccase, bio-fuel cells (BFCs) with remarkably high open circuit voltage (OCV) of up to 1.17 V[5] and with maximum power densities ($P$) ranging from 1.7[6] to even 8.4 mW cm$^{-2}$ when operated in gas breathing mode[7] and in a DET configuration were constructed[8–11]. However, their intrinsic sensitivity toward $O_2$ and high anodic potentials hampers the application of these biocatalysts in technologically relevant devices because of a fast deactivation under operating conditions[8–10]. Strategies to enhance the $O_2$ tolerance of the immobilized biocatalyst include the use of porous electrodes (blocking of $O_2$ diffusion) or the use of specific heterotrimeric membrane-bound [NiFe]-hydrogenases embedded in quinone containing lipid bilayers, as demonstrated by the Armstrong[12] and Jeuken[13] groups, respectively. The latter strategy also ensures protection from high potential deactivation even under substrate limiting conditions[13]. Also a possible reactivation of the deactivated enzyme under $H_2$ atmosphere by reduced viologen species was described[14].

An elegant and efficient way to protect hydrogenases from $O_2$ and from high potential deactivation is the incorporation of the enzyme into specifically designed low-potential viologen-modified redox polymers, as it was demonstrated for various $O_2$ sensitive hydrogenases ([NiFe][15], [FeFe][16], and [NiFeSe][17]). The low-potential redox polymer does not only act as an immobilization matrix and an electron relay between the electrode and the biocatalyst, but it also acts as a Nernst buffer system that prevents high-potential deactivation. Simultaneously, $O_2$ is eliminated by reduction at reduced viologen moieties at the polymer/electrolyte interface. The electrons for the oxygen reduction reaction (ORR) are delivered from the biocatalyst itself by transferring the electrons from $H_2$ oxidation to oxidized viologen species[15,18]. Although this strategy allows for fabrication of even membrane-free $H_2/O_2$ BFCs[15], thick films and thus rather large amounts of the biocatalyst have to be employed to ensure a discrete separation of the $H_2$ oxidation layer close to the electrode surface and the $O_2$ protection layer at the polymer/electrolyte interface (note that in thin films, the enzyme is deactivated rapidly by $O_2$; however, the reduced low-potential polymer matrix may reactivate the deactivated enzyme even in the absence of $H_2$ as it was demonstrated for a [NiFeSe] hydrogenase[17;] for a more detailed description of the protection mechanism the reader is referred to ref. [18]). Consequently, in the presence of $O_2$ parts of the turnover of the $H_2$ oxidizing catalysts are wasted for the protection of the catalytic layer and do not contribute to the $H_2$ oxidation current, i.e., electrons from the outer layer are transferred to $O_2$ and not to the electrode. This effect is illustrated in chronoamperometric experiments conducted under turnover conditions upon addition of $O_2$ to the gas feed: a reversible decrease of the oxidation current due to consumption of electrons for the ORR is observed[15–17]. Evidently, a complete protection of the active hydrogenase layer without consuming electrons from $H_2$ oxidation would be beneficial and additional protection strategies are desired.

Enzymatic $O_2$ removal systems based on an oxidase (e.g., glucose oxidase) and catalase (CAT) are well known to ensure anaerobic conditions typically in solution[19–22]. Within each catalytic cycle, $\frac{1}{2}O_2$ molecules are removed in the presence of an

electron donor, i.e., glucose in the case of glucose oxidase (GOx) (Eq. (1–3)).

$$\text{Glucose oxidase reaction : glucose} + O_2 \rightarrow \text{gluconolactone} + H_2O_2 \tag{1}$$

$$\text{Catalase reaction (disproportionation): } H_2O_2 \rightarrow H_2O + \tfrac{1}{2}O_2 \tag{2}$$

$$\text{Net reaction : glucose} + \tfrac{1}{2}O_2 \rightarrow \text{gluconolactone} + H_2O. \tag{3}$$

This enzyme cascade removes not only oxygen but also harmful $H_2O_2$ (Eq. (2)) that might be produced by an incomplete reduction of $O_2$ at the low-potential viologen mediator. Recently, we have shown that immobilization of lactate oxidase ($O_2$ removal element when lactate is present) and catalase in a redox-silent polymer matrix on top of a GOx-based sensing layer in which the biocatalyst was wired via a low potential, $O_2$-reducing toluidine blue-modified redox polymer leads to a $O_2$-insensitive GOx-based glucose-converting bioanode, which could be operated under ambient conditions[23].

In addition to the highly efficient hydrogenase-based bioanodes, stable and highly active biocatalysts with low overpotentials for $O_2$ reduction are also required for the biocathode reaction. Although bilirubin oxidase[24] and laccase[24,25] show high turnover rates for the ORR, their low intrinsic stability against halides[26–31] and deactivation by $H_2O_2$[32–34] may hamper their use in high-performance and technologically relevant BFCs.

Horseradish peroxidase (HRP) is a robust enzyme that is readily available and catalyzes the $2e^-/2H^+$ reduction of $H_2O_2$ to water[35,36]. The oxidant $H_2O_2$ can be generated in situ e.g., most frequently by GOx[37–42] or other oxidases, like pyranose oxidase[43] in the presence of glucose. Moreover, HRP can be electrically wired at extraordinary high potentials of +860 mV vs. SHE via the high potential iron-oxo complex $Fe^{IV} = O$, heme$^+$·(compound I)[35,36] in DET regime using specifically modified carbon electrodes, i.e., temperature-treated spectrographic graphite electrodes[44,45] or carbon nanotube (CNT)-modified graphite electrodes[37,38,46]. The low overpotential for the overall $4e^-/4H^+$ reduction of $O_2$ to water (via GOx and HRP) makes this enzyme cascade a promising alternative to the highly active but also sensitive multicopper oxidases for high current-density biocathodes[39–41].

Here, we report the design of a fully protected polymer multilayer-based hydrogenase bioanode combined with an oxidase/HRP biocathode for the fabrication of a $H_2$-powered BFC that consumes $H_2O_2$ as the oxidant which is generated in situ from $O_2$ by an oxidase and in the presence of glucose to keep the concentration of harmful $H_2O_2$ low. The device shows an extraordinary high OCV and remarkable current densities.

## Results

**Electrode architectures and concept.** The architecture of the proposed $O_2$-protected hydrogenase bioanode is based on a polymer multilayer architecture comprising an active $H_2$ oxidation layer in which a hydrogenase, i.e., [NiFe] hydrogenase from *Desulfovibrio vulgaris* Miyazaki F (*Dv*MF-[NiFe])[47] or [NiFeSe] hydrogenase from *Desulfovibrio vulgaris* Hildenborough (*Dv*H-[NiFeSe])[48], is wired via a low-potential viologen-modified polymer, i.e., the polymer matrix P(N₃MA-BA-GMA)-vio (poly (3-azido-propyl methacrylate-*co*-butyl acrylate-*co*-glycidyl methacrylate)-viologen, Fig. 1, right)[17], and an $O_2$ removing top layer composed of an oxidase and catalase entrapped in the redox-silent hydrophilic polymer matrix P(SS-GMA-BA)

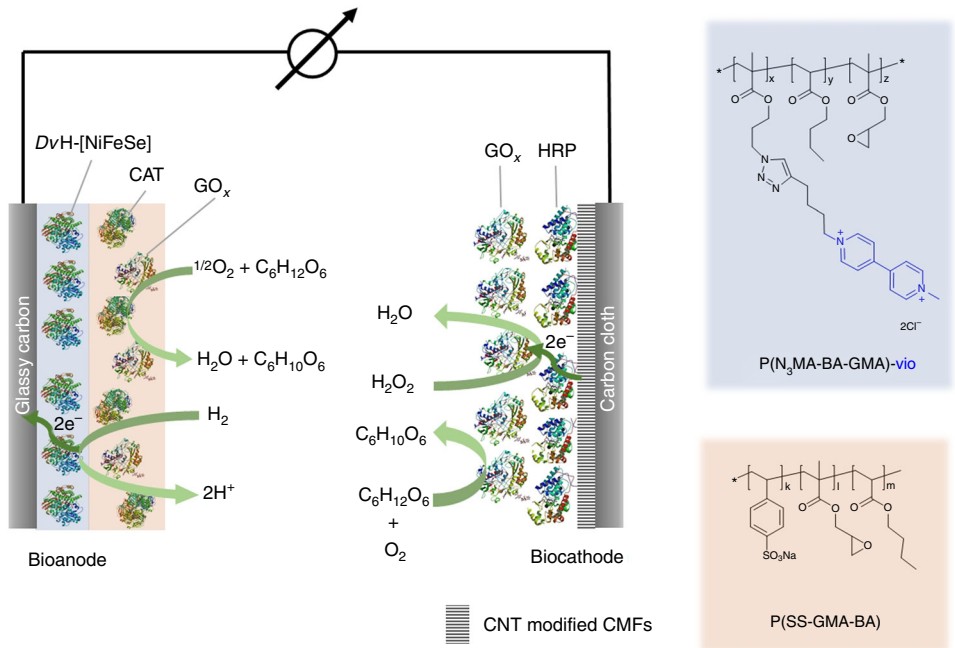

**Fig. 1** Schematic of the proposed hydrogenase/horseradish peroxidase biofuel cell. The bioanode consists of a polymer double-layer system comprising an underlying layer composed of the [NiFe] or [NiFeSe] hydrogenase from *Desulfovibrio vulgaris* Miyazaki F (*Dv*MF-[NiFe]) and from *Desulfovibrio vulgaris* Hildenborough (*Dv*H-[NiFeSe]) integrated into the viologen-modified polymer P(N$_3$MA-BA-GMA)-vio (poly(3-azido-propyl methacrylate-*co*-butyl acrylate-*co*-glycidyl methacrylate)-viologen (the viologen moieties are depicted in blue) and a protection top layer consisting of a bienzymatic system that contains an oxidase (glucose oxidase, GOx, or pyranose oxidase, Py$_2$Ox) and catalase (CAT) embedded in the redox-silent polymer P(SS-GMA-BA) (poly (4-styrene sulfonate-*co*-glycidyl methacrylate-*co*-butyl acrylate)). Oxygen is removed by converting glucose to gluconolactone by the concomitant reduction of O$_2$ to H$_2$O$_2$. The biocathode is built on a carbon cloth electrode that was modified with carbon microfibers (CMFs) that were decorated with carbon nanotubes (CNTs). The latter ensure wiring of the horseradish peroxidase (HRP) via the iron-oxo complex compound I. The cathode was first modified with pyrene butyric acid (hydrophilization) and then via a sequential drop cast process with a HRP layer followed by a top layer containing the oxidase (GOx or Py$_2$Ox) that ensures the in situ H$_2$O$_2$ formation. Note that for clarity only the combination of a *Dv*H-[NiFeSe]/GOx/CAT and GOx/HRP based electrodes are shown in the scheme

(poly(4-styrenesulfonate-*co*-glycidyl methacrylate-*co*-butyl acrylate)[49], Fig. 1, right). The bioanode configuration and the proposed protection mechanism for the active layer is illustrated in Fig. 1 (left). The oxidase/catalase O$_2$ removal system is fueled by glucose which simultaneously acts as the reactant for the in situ generation of the oxidant H$_2$O$_2$ at the oxidase/HRP-based biocathode. H$_2$O$_2$ will only be formed by the oxidase in the presence of O$_2$. Consequently, also for the proposed H$_2$/glucose fuel cell, protection of the bioanode from O$_2$ is indispensably required. Nanostructured electrodes are based on a high surface area carbon cloth that is modified with CNT-decorated carbon microfibers (CMFs), introduced for the wiring of HRP. The carbon nanostructures are a prerequisite for the productive wiring of HRP via the high-potential iron-oxo complex compound I. The proposed system provides full protection of the hydrogenases by combining O$_2$ protection by means of the bienzymatic O$_2$ removal system in the outer layer with protection from high-potential deactivation by means of the viologen-modified polymer that acts as Nernst buffer and should ensure a constant power output even in the presence of oxygen.

**Bioanode**. The efficiency of the polymer-supported oxidase/catalase O$_2$ removal system (Eq. (1–3)) for the protection of the active polymer/hydrogenase layer was first tested with bare Pt electrodes that were coated with a GOx/CAT/P(SS-GMA-BA) layer (prepared according to ref. [50]). The Pt electrode was polarized at a potential of +10 mV vs. SHE to induce ORR at the electrode surface, as can be seen from a constant cathodic current

flow (Supplementary Fig. 1a). Upon addition of glucose (50 mM) the current dropped to almost zero due to the efficient removal of O$_2$ at the interface of the protection layer. Since the polymer-bound viologen itself is a catalyst for the ORR[15], the signal of the viologen moiety in the cyclic voltammograms can be used to evaluate the efficiency of the O$_2$ removal system in the double-layer configuration. For this, pristine P(N$_3$MA-BA-GMA)-vio films were coated with a large excess of the GOx/CAT/P(SS-GMA-BA) mixture in a drop cast process to form a P(N$_3$MA-BA-GMA)-vio//P(SS-GMA-BA)/GOx/CAT double-layer system (note that for all experiments a large excess of the second layer with respect to polymer mass was used to ensure the full coverage of the active hydrogenase layer, for composition and polymer/enzyme ratios see Methods part). Cyclic voltammograms in phosphate buffer (0.1 M, pH 7.4) containing glucose (50 mM) under argon (Supplementary Fig. 1b, black dashed line) and under air (red line) show the unchanged reversible signal of the polymer-bound viologen moiety. A significant O$_2$ reduction via the low potential viologen based mediator[15], as it was observed for voltammograms recorded without glucose in solution (blue line), was absent indicating that O$_2$ is fully removed in the top layer. This also indicates that both layers do not intermix significantly during rehydration since no direct exposure of the viologen-modified polymer/hydrogenase layer to the bulk electrolyte is detected. In contrast to the experiments in quiescent solutions, slightly enhanced cathodic currents were observed due to enhanced mass transport, when a mixture of 5% O$_2$ and 95% Ar is purged through the electrolyte. Trace amounts of O$_2$ seem to reach the underlying active layer under these conditions

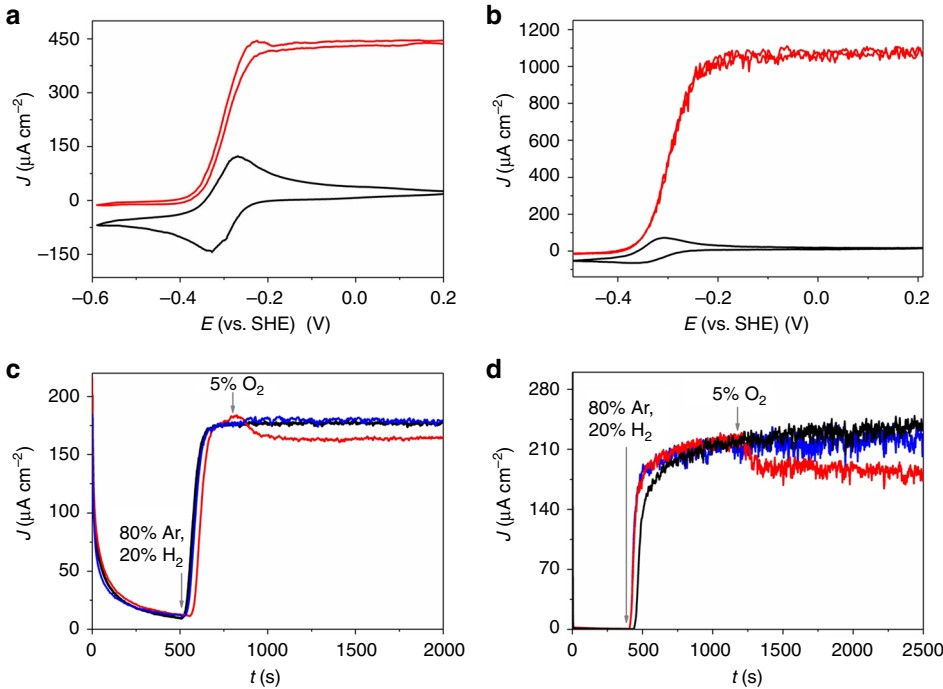

**Fig. 2** Electrochemical characterization of the bioanode. Cyclic voltammetry (**a**, **b**) and chronoamperometry (**c**, **d**) of glassy carbon electrodes modified with the polymer multilayer system comprising an underlying P(N$_3$MA-BA-GMA)-vio/hydrogenase layer (drop cast, **a**, **c**: [NiFe] hydrogenase from *Desulfovibrio vulgaris* Miyazaki F, *Dv*MF-[NiFe], **b**, **d**: [NiFeSe] hydrogenase from *Desulfovirio vulgaris* Hildenborough, *Dv*H-[NiFeSe]) and an outer protection layer based on P(SS-GMA-BA)/GOx/CAT (drop cast). Working electrolyte: 0.1 M phosphate buffer (pH 7.4), **a** and **b**: scan rate: 10 mV s$^{-1}$; black traces: 100% argon, red traces: 100% H$_2$; **c**, **d**: applied potential: +160 mV vs. standard hydrogen electrode (SHE); black traces: 80% Ar and 20% H$_2$; red traces: 75% Ar, 20% H$_2$ and 5% O$_2$ without glucose, blue traces: 75% Ar, 20% H$_2$ and 5% O$_2$ with glucose in solution (50 mM). Gray arrows in **c** and **d** indicate a change of the gas feed composition. P(N$_3$MA-BA-GMA)-vio = poly(3-azido-propyl methacrylate-*co*-butyl acrylate-*co*-glycidyl methacrylate)-viologen; P(SS-GMA-BA) = poly(4-sytyrenesulfonate-*co*-glycidyl methacrylate-*co*-butyl acrylate); GOx; glucose oxidase, CAT; catalase

(Supplementary Fig. 1c). However, under H$_2$ turnover conditions, these trace amounts of O$_2$ are reduced at the viologen moiety and hence will not significantly affect the performance of the bioanode.

Cyclic voltammograms (Fig. 2a, b and Supplementary Fig. 2a) recorded with P(N$_3$MA-BA-GMA)-vio/hydrogenase films drop cast onto glassy carbon electrodes and covered with P(SS-GMA-BA)/GOx/CAT layers under Ar and H$_2$ atmosphere show pronounced catalytic waves revealing half wave potentials (*Dv*MF-[NiFe]/P(N$_3$MA-BA-GMA)-vio: ≈ −0.27 V vs. SHE; *Dv*H-[NiFeSe]/P(N$_3$MA-BA-GMA)-vio: ≈ −0.32 V vs. SHE) that match nicely the redox potential of the redox polymer (−0.3 V vs. SHE), indicating a successful wiring of the hydrogenase in a MET regime. The current responses of the bioanodes are similar to previously reported polymer/hydrogenase-based bioanodes[15,17].

The mass transport of H$_2$ is obviously only slightly hampered by the protection layer demonstrated by the marginally higher steady state currents for the single layer system P(N$_3$MA-BA-GMA)-vio/*Dv*MF-[NiFe] under H$_2$ turnover conditions (cf. Supplementary Fig. 2a, b). The catalytic current in the double-layer system is limited by the H$_2$ concentration with a linear current increase up to 40% H$_2$ (Supplementary Fig. 2c, d). This behavior corresponds to regime III or case III following the notation described in refs. [18,51], respectively. Note that for this behavior a current decrease is expected upon O$_2$ addition if no additional protection system is used[15,18].

Chronoamperometry in the presence of glucose with *Dv*MF-[NiFe] or *Dv*H-[NiFeSe] hydrogenase-bioanodes coated with the protection layer show constant steady state H$_2$ oxidation currents under 20%H$_2$/ 80%Ar (Fig. 2c, d, black traces) and 5% O$_2$/20% H$_2$/ 80%Ar (blue lines) indicating that only insignificant numbers

of reduced viologen units are involved in the O$_2$ protection by the polymer multilayer system. In contrast, in the absence of glucose (Fig. 2c, d, red traces), the anodic current of the double-layer system decreases to a lower steady state value due to the continuous consumption of electrons from H$_2$ oxidation by O$_2$ reduction at the viologen-modified polymer matrix in analogy to electrodes coated only with a polymer/hydrogenase reaction layer[15–17].

For high H$_2$ concentrations the *Dv*H-[NiFeSe] based electrodes show higher *J*-values compared to the *Dv*MF-[NiFe] (cf. Fig. 2a, b) which is in line with our previous results[15,17]. However, for lower H$_2$ contents the difference seems to be lower (cf. Fig. 2c, d) which may be related to the fact that the currents are limited by the mass transport at these low H$_2$ levels.

Polymer multilayer modified bioanodes based on the *Dv*MF-[NiFe] hydrogenase show rather constant currents in long-term chronoamperometric experiments under H$_2$ atmosphere. Only a slight decrease of the H$_2$ oxidation current is observed over a period of 18 h (Fig. 3a, black line, for absolute currents see Supplementary Fig. 3a), indicating that the biocatalyst and the mediator (see also Supplementary Fig. 4a) are stable within the timescale of the experiment. In the absence of the protection system and in the presence of O$_2$ a rather fast linear decrease of the oxidation currents to background values within 9 h was observed since the oxygen front reaches the reaction layer within the course of the experiment (Fig. 3a, red line, Supplementary Fig. 3a, red line). This does not only lead to a deactivation of the hydrogenases but also diminishes the signal of the current amplitude of the polymer-bound viologen units in voltammo-grams recorded after the long-term experiment (Supplementary Fig. 4b) most likely due to a partial decomposition of the

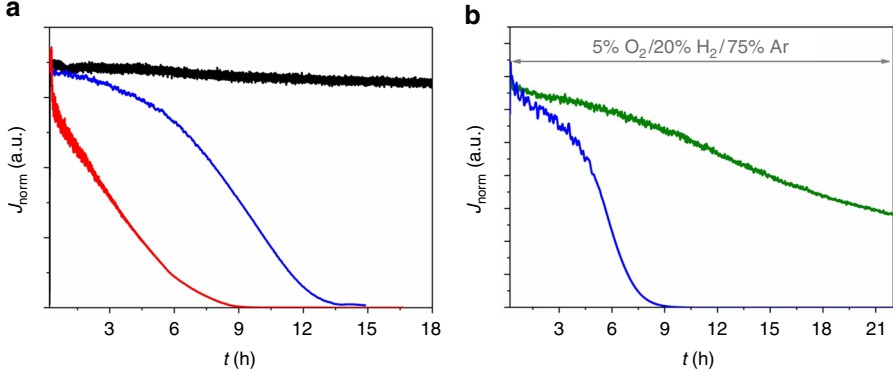

**Fig. 3** Long-term stability of the polymer multilayer-based bioanodes. Chronoamperometric long-term stability measurements of the double-layer bioanodes in 0.1 M phosphate buffer, pH 7.4 at an applied potential of +160 mV vs. standard hydrogen electrode (SHE) under turnover conditions and in the presence of $O_2$. **a** Black trace: P($N_3$MA-BA-GMA)-vio/$Dv$MF-[NiFe]//P(SS-GMA-BA)/GOx/CAT system with 20% $H_2$/80% Ar bubbling through the cell and in the absence of glucose; red trace: single P($N_3$MA-BA-GMA)-vio/$Dv$MF-[NiFe] layer, 5% $O_2$/20% $H_2$/75% Ar; blue trace: P($N_3$MA-BA-GMA)-vio/$Dv$MF-[NiFe]//P(SS-GMA-BA)/GOx/CAT system with 5% $O_2$/20% $H_2$/75% Ar bubbling through the cell and 100 mM glucose; the same enzyme batch was used for all three experiments. **b** Comparison between the protection systems based on GOx (P($N_3$MA-BA-GMA)-vio/$Dv$MF-[NiFe]//P(SS-GMA-BA)/GOx/CAT, blue line) and $Py_2$Ox (P($N_3$MA-BA-GMA)-vio/$Dv$MF-[NiFe]//P(SS-GMA-BA)/$Py_2$Ox/CAT, green line) with 5% $O_2$/20% $H_2$/75% Ar bubbling through the electrolyte and with 50 mM glucose in solution; the same enzyme batch was used for both measurements. For comparative purposes the current densities were normalized to the current flow at 100% $H_2$ ($J_{norm}$) to eliminate variations in $J$ (for absolute current values ($I$) see Supplementary Fig. 3). P($N_3$MA-BA-GMA)-vio = poly(3-azido-propyl methacrylate-$co$-butyl acrylate-$co$-glycidyl methacrylate)-viologen; P(SS-GMA-BA) = poly(4-sytyrenesulfonate-$co$-glycidyl methacrylate-$co$-butyl acrylate); GOx; glucose oxidase, CAT; catalase, $Dv$MF-[NiFe]; [NiFe] hydrogenase from *Desulfovibrio vulgaris* Miyazaki F

mediator under aerobic conditions (formation of $H_2O_2$ by the viologen).

In the case of the double-layer system in the presence of glucose and upon addition of 5% $O_2$ to the gas feed only a rather weak decrease of the $H_2$ oxidation current was observed within the first hours (0–6 h) of the experiments (Fig. 3a and Supplementary Fig. 3a, blue lines). At $t > 6$ h, a sudden current drop occurred and $H_2$ oxidation was finally stopped at $t > 13$ h. This sudden decrease of the current is striking and differs from the behavior that could be expected from the linear decay for the system without protection. Cyclic voltammograms recorded with electrodes that were transferred into a fresh electrolyte still show the less pronounced redox waves of the polymer-bound viologen units (Supplementary Fig. 4c). Hence, we conclude that the biocatalyst was irreversibly deactivated under these conditions even in the presence of the $O_2$ removal system. Moreover, reactivation via the reduced viologen moieties as it was observed for thin P($N_3$MA-BA-GMA)-vio/$Dv$H-[NiFeSe] (see ref. [17]) and P($N_3$MA-BA-GMA)-vio/$Dv$MF-[NiFe] films (Supplementary Fig. 5) deactivated by $O_2$ was not possible. Evidently, there must be an additional reason for the deactivation of the polymer entrapped hydrogenase, which is not due to the presence of $O_2$. The pH value of the buffer solution changed during catalysis from 7.4 at the beginning to 4.4 at the end of the experiment due to the formation of gluconolactone which is readily hydrolyzed to gluconic acid. In combination with the protons generated by the conversion of $H_2$ at the bioanode, the local concentration of $H^+$ within the polymer/enzyme film exceeds the buffer capacity in the solvated hydrogel and the pH drops to a critical value at which the enzyme is disintegrated.

Since Eq. (1–3) hold true for any oxidase, we exploited the use of other glucose-converting systems as protection layer, e.g., pyranose 2-oxidase ($Py_2$Ox). This particular enzyme catalyzes the oxidation of glucose to 2-dehydro-D-glucose. This rather stable and inert ketone is not prone to undergo hydrolysis in water. Consequently, the pH value of the solution does not change during catalysis as previously reported for $Py_2$Ox/CAT in solution[19]. Moreover, in situ generation of $H_2O_2$ by $Py_2$Ox was

successfully coupled to HRP[43] and is hence suitable for the fabrication of the proposed biocathode.

The performance of $Py_2$Ox/CAT to remove $O_2$ is lower than that of GOx/CAT due to the lower activity of $Py_2$Ox and $O_2$ removal is only possible up to 3% $O_2$ (Supplementary Fig. 6). However, the chronoamperometric long-term evaluation shows a constant current decay (Fig. 3b, green line). Cyclic voltammograms recorded under argon after the chronoamperometric experiment show slightly decreased signals of the viologen units indicating that also the polymer matrix, although to a lower extent when compared to the GOx-based system, is affected under these conditions (Supplementary Fig. 7). Hence, not only enzyme deactivation but also polymer degradation may contribute to the observed current decrease.

**Biocathode**. For the wiring of HRP, a carbon cloth decorated with CNT-modified CMFs was used as base electrode material[37,38]. The carbon cloth was first modified with CMFs in an Fe catalyzed chemical vapor deposition (CVD) process, followed by a second Fe catalyzed CVD process, in which the CMFs were decorated with CNTs (Supplementary Fig. 8).

The CMFs-modified carbon cloth ensures a high surface area, while the CNTs allow for the wiring of compound I within the oxidized HRP at high positive potentials[46]. Supplementary Fig. 9 shows the cyclic voltammogram of a drop cast HRP-modified CNT/CMF-carbon cloth electrode in the absence (black line) and presence of 2 mM $H_2O_2$ (red line). $H_2O_2$ reduction at compound I is counterbalanced by direct $H_2O_2$ oxidation at the carbon electrode. Thus, $H_2O_2$ reduction becomes visible at about +0.75 V vs. SHE which is more positive than the BOD catalyzed $O_2$ reduction potential of up to 0.67 V vs. SHE[31]. As compared to $O_2$ reducing biocathodes that catalyze the full $4e^-$/$4 H^+$ reduction of $O_2$ to water, the high potential of this biocathode must be paid with the loss of two electrons in the cathodic reaction. However, since the size of this electrode is easily scalable, the $2e^-$ reduction can be fully compensated using an oversized biocathode (with respect to the bioanode).

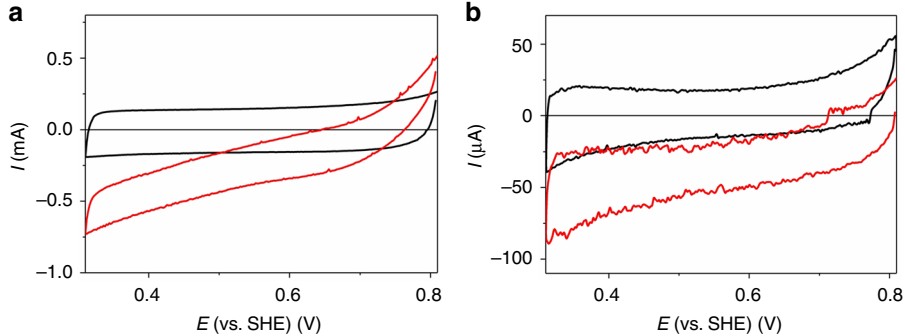

**Fig. 4** Voltammetric characterization of the biocathode. **a** Cyclic voltammograms of a glucose oxidase/horseradish peroxidase modified carbon nanotube (CNT)/carbon microfiber (CMF)-carbon cloth electrode in presence of 50 mM glucose under argon (black line) and $O_2$ (red line) atmosphere; scan rate: 5 mV s$^{-1}$. **b** Cyclic voltammograms of a pyranose oxidase/horseradish peroxidase modified CNT/CMF-carbon cloth electrode with $O_2$ bubbling through the electrolyte in absence (black line) and presence (red line) of glucose (1 mM). The geometrical surface area of the carbon cloth-based electrode was 1 cm$^2$ thus the current densities equal the absolute currents. However, it should be noted that the use of the geometrical surface is for the calculation of $J$ is a rough approximation since the electrodes reveal a rather porous 3D structure. All voltammetric scans were conducted in 0.1 M phosphate buffer, pH 7.4. SHE; standard hydrogen electrode

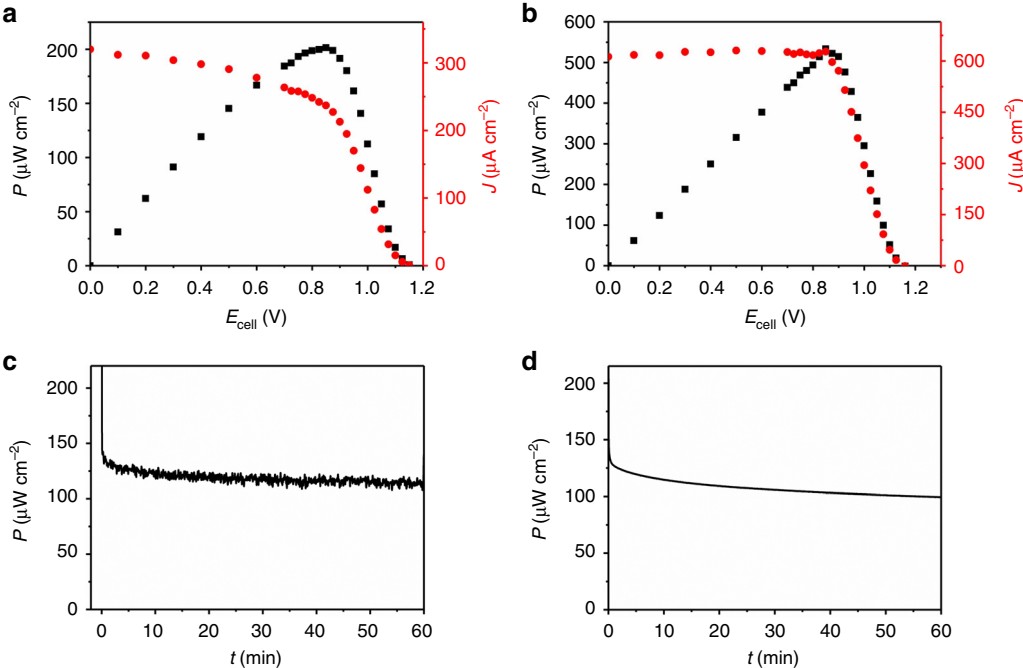

**Fig. 5** Characterization of the hydrogen/glucose biofuel cell. Performance (**a**, **b**) and stability (**c**, **d**) of the $H_2$/glucose($H_2O_2$)-powered hydrogenase ($H_2$ase)/ horseradish peroxidase (HRP)-based biofuel cells in 0.1 M phosphate buffer (pH 7.4) and in a two-compartment cell with the polymer double-layer bioanode (P($N_3$MA-BA-GMA)-vio/$H_2$ase//P(SS-GMA-BA)/$Py_2$Ox/CAT) and the $Py_2$Ox/HRP/CNT/CMF-carbon cloth biocathode. **a**: $H_2$ase = $Dv$MF-[NiFe], 1 mM glucose in both compartments; OCV = 1.15 V; $J$ = 240 μA cm$^{-2}$ and $P_{max}$ = 200 μW cm$^{-2}$ at 0.85 V. **b** $H_2$ase = $Dv$H-[NiFeSe], 3 mM glucose in both compartments. OCV = 1.15 V; $J$ = 630 μA cm$^{-2}$ and $P_{max}$ = 530 μW cm$^{-2}$ at 0.85 V. **c**, **d** Operational stability at a constant load of 0.8 V of the $Dv$MF-[NiFe] (**c**) and the $Dv$H-[NiFeSe] (**d**) based BFC; 3 mM glucose was used in the bioanode and the biocathode compartment. **a**–**d** Bioanode compartment = 97% $H_2$/3% Ar and biocathode compartment = 50% $O_2$/50% Ar. $J$ and $P$ were referenced to the geometrical surface are of the glassy carbon-based anode (0.07 cm$^2$); note that different electrodes were used for the experiments depicted in **a/c** and **b/d**, respectively. P($N_3$MA-BA-GMA)-vio = poly(3-azido-propyl methacrylate-*co*-butyl acrylate-*co*-glycidyl methacrylate)-viologen; P(SS-GMA-BA) = poly(4-sytyrenesulfonate-*co*-glycidyl methacrylate-*co*-butyl acrylate); $Py_2$Ox; pyranose oxidase, CAT; catalase, $Dv$MF-[NiFe]; [NiFe] hydrogenase from *Desulfovibrio vulgaris* Miyazaki F, $Dv$H-[NiFeSe]; [NiFeSe] hydrogenase from *Desulfovibrio vulgaris* Hildenborough

The immobilization of the oxidase for the in situ generation of $H_2O_2$ was performed in a second drop cast process on top of the HRP layer. Cyclic voltammograms of GOx/HRP (A) or $Py_2$Ox/HRP (B) modified CNT/CMF-carbon cloth electrodes recorded under argon (black line) and under $O_2$ (red line) in 0.1 M PB (pH 7.4) containing 50 mM glucose are shown in Fig. 4. Obviously, both enzymes are able to produce substantial amounts of the

oxidant $H_2O_2$ as indicated by the pronounced catalytic waves under turnover conditions (red lines, note that the current response of the carbon cloth-based electrodes is similar to our previously described systems based on CNT/CMF modified carbon rods[37,38]). While the absolute currents are lower than those obtained for electrodes equipped with only HRP in the presence of 2 mM $H_2O_2$, the potential at which visible $H_2O_2$

reduction occurs does not significantly change for the oxidase/ HRP system. Moreover, by varying the glucose concentration, the current output at the biocathode can be controlled and adjusted to that of the bioanode (Supplementary Fig. 10a). In addition, chronoamperometric experiments conducted with the GOx/HRP system does not lose its activity over a period of 6 h (Supplementary Fig. 10b).

**Biofuel cell**. The performance of a BFC composed of the double-layer-protected hydrogenase bioanode and the oxidase/HRP biocathode was evaluated in a two-compartment cell separated by a Nafion membrane. A BFC comprising a $Dv$MF-[NiFe]/P (N$_3$MA-BA-GMA)-vio//Py$_2$Ox/CAT/P(SS-GMA-BA) bioanode and a Py$_2$Ox/HRP/CNT-CMF-carbon cloth biocathode exhibits an OCV of about 1.15 V (Fig. 5a), which is significantly larger than the OCV of reported polymer-based [NiFe] (0.95 V)[15] or [FeFe] (1.08 V)[16] hydrogenase/BOD BFCs. For the $Dv$H-[NiFeSe] bioanode an identical OCV was measured (Fig. 5b). To ensure limiting conditions for the bioanode, an oversized biocathode was used in combination with appropriate glucose concentrations between 1 to 3 mM (same concentration in both compartments was used). Cyclic voltammograms measured with the individual half cells show that the absolute oxidation currents of the bioanodes (Supplementary Fig.11a, c) are indeed smaller than the absolute reduction current at the biocathode (Supplementary Figs. 11b–d) which assures the bioanode being the limiting electrode in all experiments. Even under these harsh conditions the $Dv$MF-[NiFe] bioanode is stable and the corresponding BFC shows a maximum power density of 200 µW cm$^{-2}$ at 0.85 V (referenced to the surface area of the anode). The $Dv$H-[NiFeSe] system shows even higher current densities and a maximum power density of 530 µW cm$^{-2}$ at 0.85 V. Cyclic voltammograms (Supplementary Fig. 11, dashed lines) of the individual half cells measured after the BFC test show pronounced catalytic currents and the signals of the viologen redox couple indicating that no deactivation occurs within the timescale of the experiment. Interestingly, the catalytic currents for the cathodes are much higher after the experiment, most likely due to an accumulation of H$_2$O$_2$ in the porous electrode structure during the BFC test. Different BFCs prepared with the same enzyme batch show similar values (Supplementary Fig. 12).

The operational stability of the two-compartment BFC was evaluated at a constant load. Figure 5c and d shows the power output over 1 h at 0.8 V for the $Dv$MF-[NiFe] and $Dv$H-[NiFeSe] based BFC, respectively. Only a slight drop of the initial current from 123 µW cm$^{-2}$ ($t = 10$ min) to 113 µW cm$^{-2}$ ($t = 60$ min) for the [NiFe] based system and from 114 µW cm$^{-2}$ ($t = 10$ min) to 100 µW cm$^{-2}$ ($t = 60$ min) for the [NiFeSe] based system was observed. The operational stability is similar to a high current-density hydrogen BFC that is working in a DET regime using a thermostable and O$_2$ tolerant hydrgenase[52].

The long-term stability at a constant load of 0.8 V was evaluated in a two-compartment cell configuration with a $Dv$H-[NiFeSe] based BFC (Supplementary Fig. 15a). The power output reaches 50% of its initial value after ≈ 10 h. After 20 h still 25% of the initial power output could be detected. A closer voltammetric inspection of the individual electrodes (Supplementary Fig. 15b and c) revealed that the turnover currents at the Py$_2$Ox/HRP based biocathode even after the addition of further glucose to the solution significantly decreased and are close to the background current. While the currents of the viologen units remain constant before and after the long-term test. The H$_2$ oxidation currents decreased by about 30% indicating that parts of the hydrogenase were disintegrated at the bioanode. The results clearly demonstrate that the biocathode is the limiting electrode with respect to

durability and that the bioanode shows the envisaged high stability even under operational conditions.

Since the protection system is able to remove significant amounts of detrimental O$_2$ at the anode, also a membrane-free single-compartment cell was evaluated using a bioanode based on the highly active $Dv$H-[NiFeSe] hydrogenase (Supplementary Fig. 13). To avoid critical H$_2$/O$_2$ ratios, the oxygen content in this system was kept constant at 3%, which results in lower currents for the cathode reaction. However, the anode is still the limiting electrode as shown in the cyclic voltammograms of the half cells (Supplementary Fig. 14, red solid lines). The corresponding single-compartment BFC reveals a $P_{max}$ of 160 µW cm$^{-2}$ at 0.85 V. The OCV was 1.15 V and thus identical to the two-compartment cell.

Comparison of the cyclic voltammograms of the biocathode under turnover conditions after BFC evaluation show that in the single-compartment configuration the activity of the HRP biocathode was almost completely lost (Supplementary Fig. 14, dashed lines). The bioanode shows a similar current response before and after the BFC test (Supplementary Fig. 14a). The lower power output and lower stability of the one-compartment cell compared to the two-compartment system might be a result of unwanted side reactions that are induced by intermixing all substrates/products in the cell which may lead to a chemical disintegration of especially the biocathode. For the hydrogenase/ BOD based systems where no glucose and H$_2$O$_2$ is present such effects were not observed. This reflects a clear limitation of the system reported here. However, as indicated by voltammograms measured before and after the BFC test, which are almost identical, the bienzymatic protection system in combination with the viologen-modified polymer matrix shows the envisaged full protection of the hydrogenase-based bioanode.

In contrast, when an inactivated catalase was used for the preparation of the double-layer bioanode ([NiFeSe] based system), the evaluation of the fully assembled fuel cell was not possible due to the fast damage of the bioanode with the current dropping to zero within a short time (Supplementary Fig. 16a). Comparison of cyclic voltammograms recorded before and after the BFC test unambiguously shows that not only the hydrogenase itself but also the redox polymer matrix was destroyed due to the local high H$_2$O$_2$ concentrations during testing. This is evident from the complete loss in H$_2$ oxidation currents as well as the redox transition of the polymer-bound viologen moieties after the BFC test (Supplementary Fig. 17a, b). Moreover, also the biocathode took severe damage and its catalytic response was significantly reduced after the BFC evaluation (Supplementary Fig. 17c, d). This deactivation may be related to a voltage drop at the cathode caused by a change of the OCV due to the continuous disintegration of the bioanode leading to the destruction of the biocatalyst by applying extreme potentials. However, the exact nature of this deactivation of the biocathode remains unclear. Most importantly, the developed bioanode with full protection remains active even under harsh conditions (see, e.g., Supplementary Fig. 14a).

## Discussion

The bienzymatic oxidase/catalase second layer immobilized in a redox-silent hydrophilic polymer matrix ensures an efficient protection of the highly O$_2$-sensitive $Dv$MF-[NiFe] and $Dv$H-[NiFeSe] hydrogenases. Protection from O$_2$ is completely covered by the bienzymatic system and no electrons from the H$_2$ oxidation are necessary to be wasted for additional O$_2$ reduction at the polymer-bound viologen moieties. Thus, the bioanode provides the envisaged constant current response even in the presence of O$_2$. Moreover, using Py$_2$Ox instead of GOx in the O$_2$ removal

system provides a stable pH value even during long-term runs. However, the higher stability is counterbalanced by a lower protection efficiency of the $Py_2Ox$ system (3% $O_2$) compared to the GOx-based system (5% $O_2$).

Even though $O_2$ protection is decoupled from the polymer/hydrogenase reaction layer, the viologen-modified redox polymer is important not only as a fast electron-transfer redox mediator for the entrapped hydrogenase, but also since its low potential furthermore protects the hydrogenase from high-potential deactivation and ensures removal of $O_2$ traces that may break through the bienzymatic $O_2$ removal layer. Evidently, by incorporation of sensitive hydrogenases in a viologen-modified polymer layer in combination with a top layer comprising the bienzymatic $O_2$ removal system the proposed bioanode configuration ensures a complete protection of the hydrogenase even under continuous operation.

The low potential of the viologen-modified polymer allows for $H_2$ oxidation to occur at low overpotentials, which in combination with a biocathode based on HRP that is wired via the high potential iron-oxo complex compound I ensures a benchmark OCV of around 1.15 V for a polymer-based $H_2$/glucose($H_2O_2$) BFC.

In the presence of $H_2$ and glucose, which simultaneously acts as reactant for the $O_2$ removal layer at the bioanode and as enzyme substrate for the formation of the oxidant $H_2O_2$ at the biocathode, a maximum power density of about 0.5 mW cm$^{-2}$ at a remarkably high cell voltage of $\approx 0.85$ V was achieved when the highly active [NiFeSe] hydrogenase from *Desulfovibrio vulgaris* Hildenborough was used. The $D$vH-[NiFeSe] containing BFC reveals a performance that outperforms a previously reported polymer/hydrogenase systems using a BOD based biocathode[15]. In contrast, the cell is inferior to previously described $H_2$/$O_2$ hydrogenase-based BFCs operated in a DET mode[6–8], which are able to reach power densities of up to 8.4 mW cm$^{-2}$ in a gas breathing configuration[7]. However, these systems lack active protection against $O_2$.

In conclusion, we developed a polymer double-layer air-stable hydrogenase bioanode that was successfully incorporated into a $H_2$/glucose($H_2O_2$) BFC with an extraordinary high OCV that sets a benchmark for redox polymer-based BFCs. The proposed bienzymatic-protection system may be transposed to other $O_2$-sensitive biocatalysts or artificial catalysts. Moreover, the integration of additional or other enzymes into the protection layer may allow the establishment of new shielding systems that are able to remove not only $O_2$ but also other detrimental interferences that may attack sensitive catalysts of any type and nature.

## Methods

**Chemicals and materials**. All chemicals were purchased from Sigma-Aldrich, Alfa-Aesar, Acros Organics, VWR, Merck, J.T. Baker or Deutero and were used as received except otherwise noted. All chemicals were of reagent or higher grade. Gases used for the preparation of the CMF/CNT modified carbon cloth revealed a purity of N6.0. All aqueous solutions were prepared with Milli-Q water from a water purification system (Millipore).

**Polymers**. The synthesis and characterization of the viologen-modified polymer P(N$_3$MA-BA-GMA)-vio (Fig. 1) was described elsewhere[17]. The actual composition of the polymer backbone was $x = 71$ mol%, $y = 20$ mol%, $z = 9$ mol% (determined by NMR spectroscopy)[17]. The synthesis of the hydrophilic and redox-silent polymer P(SS-GMA-BA) (Fig. 1) was described in ref. [49], the nominal concentration of the individual co-monomers was $k = 50$ mol%, $l = 30$ mol%, and $m = 20$ mol%. Note that due to overlapping signals in the NMR spectrum of P(SS-GMA-BA) the actual composition cannot be determined via the integral ratios. However, all characteristic signals could be identified, and their chemical shifts are consistent with the proposed molecular structure[49]. For electrode modifications the polymers were dissolved in water with concentrations of 8 mg mL$^{-1}$ for P(N$_3$MA-BA-GMA)-vio and 60 mg mL$^{-1}$ for P(SS-GMA-BA). The polymer stock solutions were stored at room temperature.

**Enzymes**. Glucose oxidase (GOx) from *Aspergillus niger* (Type X-S, lyophilized powder, 100,000-250,000 U g$^{-1}$ solid), catalase (CAT) from bovine liver (lyophilized powder, 2,000-5,000 U mg$^{-1}$ protein), pyranose oxidase (Py$_2$Ox) from *Coriolus* sp. (recombinant, expressed in *E. coli*, $\geq 2.7$ U mg$^{-1}$ solid) and HRP (Type VI-A, lyophilized powder, salt free, 325 U mg$^{-1}$) were purchased from Sigma-Aldrich and stored at $-20$ °C. Stock solutions with enzyme concentrations of 10 mg mL$^{-1}$ were prepared in phosphate buffer (0.1 M, pH 7.4) and stored at 4 °C.

The [NiFe] hydrogenase from *Desulfovibrio vulgaris* Miyazaki F ($D$vF-[NiFe]) was isolated and purified according to ref. [53,54]. It was stored in MES buffer at pH 6.8 at $-20$ °C with a concentration of 200 µM. The $H_2$ production activity of this type of hydrogenase was determined to be 610 U mg$^{-1}$[54]. However, the activities of the individual batches varied and electrodes showed different activities toward the oxidation of $H_2$ ($\pm 50$%, with respect to the average value). The recombinant form of the [NiFeSe] hydrogenase from *Desulfovibrio vulgaris* Hildenborough ($D$vH-[NiFeSe]) was isolated and purified as described previously in ref. [48]. The activity for $H_2$ formation was estimated to be 4000–5700 U mg$^{-1}$, depending on the batch with variations of up to 320 U mg$^{-1}$ for each single batch. The activity of modified electrodes varied by $\pm 90$% (with respect to the average value). The enzyme was stored at $-80$ °C in 20 mM Tris-HCl at pH 7.6 with a concentration of 14–15 µg µL$^{-1}$ (159–170 µM). Note that rather high concentrations of the enzymes were used to achieve high biocatalyst loading on the electrode surface while keeping the volumes use the drop cast process rather small to minimize drying time and facilitate the modification of the 3 mm electrodes.

Although, the activity of the individual electrodes toward $H_2$ oxidation varied for different enzyme batches which lead to a variation in the absolute current response, all electrodes showed the same trend under different conditions thus allowing a qualitative comparison of the individual systems.

**Electrochemical experiments**. All voltammetric and chronoamperometric experiments were conducted in a standard three electrode gas-tight electrochemical cell under Ar atmosphere or Ar/$H_2$/$O_2$ mixtures at room temperature using a Reference 600 (Gamry Instruments), an Autolab PGSTAT12 (Metrohm-Autolab) or an Autolab FRA 2 Type III (Metrohm-Autolab) potentiostat. The counter electrode was a Pt wire. As reference electrode a Ag/AgCl/3 M KCl system was used. All potentials are rescaled with respect to the standard hydrogen electrode (SHE) which is $+210$ mV more negative than the Ag/AgCl/3 M KCl system. For the fabrication of the hydrogenase-bioanodes glassy carbon electrodes with a nominal diameter of 3 mm and thus a geometrical surface area of 0.07 cm$^2$ were used. Phosphate buffer (PB, 100 mM, pH 7.4) served as working electrolyte for all measurements. For measurements with different Ar/$H_2$/$O_2$ ratios three separated mass flow controllers were used to control the gas flow of the individual gases. The mass flow controllers were directly connected to the cell via a single gas-inlet. The gases were pre-mixed and then purged through the electrolyte solution and no additional stirring or rotation of the electrode was applied. Prior to each experiment, the glassy carbon working electrodes were subsequently polished using several alumina/water suspensions with decrease particle grain size (going from 1 µm, via 0.3 µm to 0.05 µm), each with $\approx 1$ min. After each polishing step the electrodes were thoroughly rinsed with water. After final polishing step the electrodes were again washed with water and dried in an argon stream to remove any dust particles.

BFC tests were either conducted in a one- or two-compartment cell by using a hydrogenase bioanode and a GOx/HRP or Py$_2$Ox/HRP biocathode. As separator in the two-compartment cell a Nafion membrane was used. Both compartments revealed a gas outlet to avoid extensive mixing of $H_2$ and $O_2$. $H_2$ and $O_2$ were bubbled through the corresponding compartment. For the one-compartment cell, a gas mixture of 90% $H_2$/5% $O_2$/5% Ar was used. Power curves were obtained by stepped potential chronoamperometric experiments. This technique minimizes contributions from background currents. Steady state currents were used to calculate the power value of the corresponding BFCs. To ensure anode limiting conditions, the oxygen flow in the cathodic compartment of the two-compartment cell was adjusted to achieve absolute currents that were significantly higher than those measured at the anode. Since for all experiments the bioanode acted as the limiting electrode, a surface area of 0.07 cm$^2$ was used to calculate current and power densities.

**Fabrication of CNT/CMF-modified carbon cloth electrodes**. For fabrication of the biocathodes carbon cloth modified with CMFs which were decorated with CNTs were used. These high surface area electrodes were prepared according to protocols described in refs. [37,38]. For the growth of the CMFs, first an iron-based catalyst was deposited onto the base electrode material in an electrochemical process. The catalyst was deposited from an aqueous solution containing FeSO$_4$·7H$_2$O (0.50 M) and MgSO$_4$·7H$_2$O (0.58 M) by applying a multi-potential pulse sequence with $n$(0 V/10 s; $-0.6$ V/2 s) (with $n = 20$, both potentials vs. Ag/AgCl/3 M KCl). The carbon cloth electrodes were pre-wetted with ethanol and water and then directly immersed into the electrolyte solution containing the iron precursor. As counter electrode a Pt gauze and as reference electrode a Ag/AgCl/3 M KCl system was used. The same procedure was used for the deposition of the iron catalyst on the CMFs for the growth of the CNTs. After CMF/CNT deposition the electrodes were thoroughly washed with water.

For growth of CMFs the iron catalyst decorated electrode material was placed in a quartz crucible and mounted in a quartz tube (100 cm length with an inner diameter of 3 cm). The tube was placed in a triple-zone tube furnace (Carbolite) and connected to a computer-controlled mass flow controller system. The sample was heated to 850 °C with an increase of 20 °C min$^{-1}$ and then to 1150 °C with 3 °C min$^{-1}$ in a H$_2$/CH$_4$ mixture (2.3:1). After 20 min at 1150 °C the temperature was lowered to 950 °C under pure H$_2$ atmosphere. Then, the sample was allowed to cool down to room temperature under 100% He atmosphere.

For CNT growth on the CMFs, the same setup was used. First the temperature was raised to 700 °C (10 °C min$^{-1}$) under a H$_2$/He atmosphere (1:1). The temperature was increased to 760 °C (10 °C min$^{-1}$) and kept constant for 20 min in a gas mixture if H$_2$/C$_2$H$_4$ (2.3:1). The tube was flushed with 100% H$_2$ for additional 30 min and finally the sample cooled down to room temperature in He atmosphere.

The electrodes were characterized by means of scanning electron microscopy. Supplementary Fig. 8a shows the bare carbon cloth. Supplementary Fig. 8b depicts the CMF modified carbon cloth with a diameter of the CMFs of ≈ 0.5 μM randomly distributed on the carbon cloth fibers. The CNT decorated material reveals a porous structure as shown in Supplementary Fig. 8c.

**Electrode modification**. Bioanodes: for the preparation of the hydrogenase-bioanodes glassy carbon electrodes (diameter = 3 mm) were first modified with a hydrogenase/P(N$_3$MA-BA-GMA)-vio layer by pre-mixing an aqueous solution of the polymer (8 mg mL$^{-1}$ in water, 3 μL), the enzyme solution (DvMF-[NiFe]: 200 μM in MES pH 6.8, 0.5 μL; DvH-[NiFeSe]: 159–170 μM in Tris-HCL at pH 7.8, 0.5 μL) and Tris-HCl-buffer (0.1 M, pH 8.6, 3 μL). Electrodes were then modified by drop casting (1.3 μL polymer/enzyme mixture) and dried at 4 °C for ≈4 h. The enzymatic-protection layer comprising GOx and CAT embedded in P(SS-GMA-BA) was added by drop coating on top of the dried polymer/hydrogenase layer. For this, the polymer P(SS-GMA-BA) (60 mg mL$^{-1}$ in water, 10 μL), GOx (10 mg mL$^{-1}$ in 0.1 M PB, pH 7.4, 10 μL) or Py$_2$Ox (10 mg mL$^{-1}$ in 0.1 M PB, pH 7.4, 10 μL), CAT (10 mg mL$^{-1}$ in 0.1 M PB, pH 7.4, 10 μL) and the bifunctional crosslinker 2,2′-(ethylenedioxy)diethylamine (1:37 in water, 1 μL) was pre-mixed and drop coated (31 μL) onto the modified electrode in a way that the underlying reaction layer was homogeneously covered. Finally, the modified electrodes were dried under ambient conditions over night.

For measurements with only the hydrogenase/polymer or GOx/CAT/polymer bioanodes, respectively, the electrodes were prepared using the same polymer/enzyme/crosslinker ratios.

Biocathodes: Oxidase/HRP-modified carbon cloth electrodes were prepared in a sequential process starting with the modification of the CNT/CMF decorated carbon cloth (geometrical surface area = 1 cm$^2$) with pyrene butyric acid (PBA). For this, the electrode was immersed for 1 h into dimethylformamide containing PBA at a concentration of 20 mg mL$^{-1}$. The PBA modified electrode was thoroughly rinsed with dimethylformamide and water. In a second step the surface was modified with HRP (10 mg mL$^{-1}$, 20 μL) by drop casting. After drying, the second enzyme layer, i.e. GOx or Py$_2$Ox, was added (10 mg mL$^{-1}$, 20 μL). The modified electrodes were dried and rinsed with water to remove any loosely bound enzyme.

## Data availability

The data that support the findings of this study are available from the corresponding author upon reasonable request.

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

## Acknowledgements
The authors thank Dr. Ines Ruff (Thermo Fisher Scientific) for FTIR measurements and Sarra Zerria for preliminary polymer syntheses. We are grateful to Nina Breuer and Patricia Malkowski for the preparation of the [NiFe] hydrogenase from *Dv*MF. The work was supported by the Deutsche Forschungsgemeinschaft in the framework of the Cluster of Excellence RESOLV (EXC 1069) and DIP (LU 315/17-1/2), by the DFG-ANR within the projects SHIELD PL746/2-1 and ANR-15-CE05-0020, and by the Fundação para a Ciência e Tecnologia (Portugal) (Grants UID/Multi/04551/2013, LISBOA-01-0145-FEDER-007660 (cofunded by FCT/MCTES and FEDER funds through COMPETE2020), PTDC/BBB-BEP/2885/2014 and PhD fellowship SFRH/BD/100314/2014.

## Author contributions
A.R. and W.S. conceived the study. A.R. prepared and characterized all polymers. S.Z. and I.A.C.P purified, characterized and provided the NiFeSe hydrogenase. W.L. provided the NiFe hydrogenase. J.S. prepared the bioanodes and performed most of the electrochemical measurements. N.M. and F.C. prepared and characterized the biocathodes. A.R., J.S., and W.S. analyzed the results and interpreted the data. All authors contributed to writing of the manuscript.

## Additional information

**Competing interests:** The authors declare no competing interests.

