## [Peer Review File · Nature Communications]

Reviewers' comments:

Reviewer #1 (Remarks to the Author):

My previous comments to the original version of this manuscript remain valid for the revised version. For enzymatic biofuel cell, the long-term discharge of the cell is more important than that of the single electrode (the test method can be referenced in *ASC Catal.*, 2017, 7, 4408; *Chem. Commun.*, 2015, 51, 7447) and the authors do not offer the experimental data. Besides, the authors claim that the bioanode and biocathode take severe damage under one chamber condition and do not clearly clarify which factors have influence on bioanode or biocathode. Finally, the superiority of bioanode in the protection of oxygen damage is not highlighted in the BFC system. In the revised manuscript, my suggestions and questions are not well considered and answered. Thus I think the revised manuscript is not suitable to publish in *Nat. Commun.*.

Reviewer #2 (Remarks to the Author):

Most of my previous questions have been answered. Therefore, I would accept its publication in *Nature Communications* after a minor revision. I still have one comment which should be further clarified and discussed. It is still not clear to me which of the bioanode and the biocathode is limiting the power of the fuel cell. Looking at the current density of the cathode, the peroxidase electrode might be limiting the power of the fuel cell when comparing normalized current density. This is the reason why I asked in the previous review of this manuscript to give the power density referenced to the average surface area of both the cathode and the anode since the limiting electrode might be the peroxidase-based cathode. Could the authors clarify this issue? Also, provide a photograph or a more precise scheme of the fuel cell would help the reader to understand the concept of this fuel cell on a more technological level.

Reviewer #3 (Remarks to the Author):

This work reports the construction of a layered hydrogenase-based bioanode, which aims to protect sensitive hydrogenases both from potential-induced inactivation and O₂ damage. A viologen-based polymer is used as a redox mediator matrix and as a buffer layer to protect hydrogenase from potential inactivation. A second layer made of glucose oxidase and catalase entrapped in a redox silent polymer serves to consume O₂. This strategy aims to overcome the issues raised by the use of the viologen-based polymer, i.e. requirement of a thick film, and waste of electrons for O₂ reduction. Stability of the bioanode over around 1 hour is demonstrated using electrochemistry on two different hydrogenases.

A new issue however comes from the formation of gluconic acid which decreases the electrolyte pH to such an extent that the hydrogenase is denatured. The use of another oxidase for glucose oxidation with production of a neutral ketone is proposed, which however does not allow to efficiently remove O₂. Nevertheless, the experiments are globally well conducted and will interest scientists working on enzyme bioelectrochemistry.

One main concern is the poor stability of the bioanode in the absence of the second protective layer compared to previous works by the same group (a few hours against one month in *Nature Chem.* 2014), which needs to be discussed in the present work.

The so-built bioanode is coupled to a biocathode based on horseradish peroxidase which reduces H₂O₂ formed in situ by co-immobilized glucose oxidase. The biofuel cell delivers power densities in the range of 500 μ W/cm² at 0.85 V. Although this power output is not very impressive, such a biofuel cell concept is new and may open new avenue if the issue linked to strong acidification of the electrolyte is solved in the future.

The concept of the protective bilayer bioelectrode was recently developed for glucose oxidase, and the viologen-redox polymer was applied to various hydrogenases. But this is the first time such a construction is used to provide a full protection to hydrogenase. In that sense, this work is innovative, and worth being published, after the following points are addressed.

Points to be addressed before publication:

- A discussion is required about the spatial organization of the two layers. How to insure stable layered structure and no fusion of the two layers once rehydrated ?
- As discussed in a previous paper, one important point in order to understand and control the protective effect of the bienzymatic layer is the thickness of the different layers. Information and discussion about this parameter is required.
- Different hydrogenase batches have been used which could explain some discrepancies in the results. Gels showing the purity of these different batches as well as specific activity are then required.

Reviewer #4 (Remarks to the Author):

In this work by Ruff et al., the construction of novel multilayer redox-polymer/biocatalyst electrode surfaces are described with perceived application in fuel cells, in which hydrogen is reduced at the anode by a biocatalyst, hydrogenase, which is well known to be deactivated by oxygen. The experiments are very carefully performed and described in very good detail, supporting the conclusions by the authors. The main advance of this work, compared to the previous state of the art, lies in the fact that the hydrogenase-modified anode is protected from oxygen damage. The authors have previously described protection mechanisms against oxygen damage that relied on relatively thick polymer layers on the anode. In this work, however, a different strategy is employed that relies on an oxygen reducing protective enzyme layer, which deposited on top of the hydrogenase layer. The main advance of the new strategy – to my opinion – is that thinner hydrogenase-polymer layers can be used potentially reducing the amount of hydrogenase catalyst.

There are couple of main issues with the strategy employed that need to be addressed prior to publication.

1) The authors describe their fuel cell as a "hydrogen/hydrogen peroxide fuel cell". However, their fuel cells is NOT a hydrogen/hydrogen peroxide fuel cell, but a hydrogen/glucose fuel cell. The cell as described by the authors would not be able to be fuelled directly by hydrogen peroxide as the catalase on the anode would consume the hydrogen peroxide in a loss making side reaction. Furthermore, the authors have not shown that the bioanode can withstand hydrogen peroxide at, for instance, 2 mM concentration. The fact that this is hydrogen/glucose fuel cells needs to be made clear in the title, abstract and main manuscript.

2) Previous enzyme-based fuel cells described in literature are often hydrogen/oxygen fuel cells. In hydrogen/oxygen fuel cell it is perceivable that oxygen leaks to the anode and hence oxygen can inactive the hydrogenase catalyst, and thus oxygen damage needs to be protected against. However, in the work by Ruff et al., a hydrogen/glucose fuel cell has been constructed. What is the author's argument that the anode needs to be protected against oxygen in a hydrogen/glucose fuel cell?

3) The protection layer on the anode consumes glucose, but glucose is also its fuel at the cathode. How much glucose is consumed at the anode has not been measured by the authors. An important aspect of a fuel cell is the Faradaic yield and this needs to be reported by the authors, especially seen the aspect just described.

Minor issues:

a) In Line 150, the authors write that "a large excess of the Gox/CAT/P(SS-GMA-BA)" layer was used in the preparation of the data shown in Supplementary Data Figure 1. Can the authors clarify if this large excess was used in all experiments or only for Supplementary Data Figure 1.

b) In Line 200-202, the authors note that the "for lower hydrogen contents the difference seems to be lower" and ascribe this to "change in film swelling". However, from Figure Supplementary Data 2C and 2D, it is clear that the current is mass transport limited. I might have misunderstood, but is the smaller difference between Figure 2C and 2D not simply due to the fact that both currents are limited by mass transport of hydrogen and thus very similar in both cases, independent on the biocatalyst used?

c) In line 234, the authors suggest the "protons generated by the conversion of hydrogen" could contribute to the lowering of the electrolyte pH. However, in this case a three electrode cell has been used with a Pt counter electrode (line 444), thus for every hydrogen oxidised at the working electrode, either (half a) oxygen is reduced at Pt electrode or two protons reduced to hydrogen. Would the authors agree that the Pt counter electrode is likely to prevent pH changes due to hydrogen oxidation by hydrogenase?

d) In Line 265-267, the authors claim an "almost identical intensity indicating that that the polymer matrix is not affected under these conditions". However, in Supplementary Figure 7B, about 20-30% of the viologen signal seems to have disappeared while the hydrogen oxidation current is reduced by ~60%. As there is no easy way to correlate the "density" of viologen in the polymer to the electronic coupling of the hydrogenase, it cannot be excluded that damage the viologen in part contributes to the reducing in current.

e) In Line 370-272, can the authors acknowledge that the drawback of Py2Ox is that is less able to prevent oxygen damage and hence only protects up to 3% oxygen.

f) Line 444, could the authors give a reference in which the standard protocol for polishing is described. It is know that details of polishing can have large effect on electrochemical behaviour.

Note: response to reviewers are written in red; changes in the manuscript are highlighted in yellow, text of the manuscript is given in blue.

Reviewer #1 (Remarks to the Author):

My previous comments to the original version of this manuscript remain valid for the revised version. For enzymatic biofuel cell, the long-term discharge of the cell is more important than that of the single electrode (the test method can be referenced in ASC Catal., 2017, 7, 4408; Chem. Commun., 2015, 51, 7447) and the authors do not offer the experimental data.

We agree that the operational stability is an important characteristic for the proposed novel biofuel cell system. Following the suggestion of the reviewer, we conducted discharge experiments at a constant load with the best performing fuel cell configuration, i.e. the two-compartment cell system in accordance to the procedures in the references mentioned by the reviewer. The results are shown in Figures 5C and D. Both cells show a stable power output over a period of 60 min (as compared to only 17 min for the hydrogenase biofuel cell described in Chem. Commun., 2015, 51, 7447).

Moreover, the long-term stability under continuous operation and a constant load of 0.8 V was now evaluated over a period of 20 h. The NiFeSe based BFC shows 50 % power drop after ≈ 10 h and still remains 25 % of its initial power output after 20 h. See, new Supplementary Figure 15.

We added these additional results and the corresponding discussion to the main text:

The operational stability of the two-compartment biofuel cell was evaluated at a constant load. Figure 5C and D shows the power output over 1 h at 0.8 V for the *DvMF*-[NiFe] and *DvH*-[NiFeSe] based BFC, respectively. Only a slight drop of the initial current from $123 \mu\text{W cm}^{-2}$ ($t = 10$ min) to $113 \mu\text{W cm}^{-2}$ ($t = 60$ min) for the [NiFe] based system and from $114 \mu\text{W cm}^{-2}$ ($t = 10$ min) to $100 \mu\text{W cm}^{-2}$ ($t = 60$ min) for the [NiFeSe] based system was observed. The operational stability is similar to other hydrogenase based air-breathing biofuel cells.⁵¹

The long-term stability at a constant load of 0.8 V was evaluated in a two-compartment cell configuration with a *DvH*-[NiFeSe] based BFC (Supplementary Figure 15A). The power output reaches 50 % of its initial value after ≈ 10 h. After 20 h still 25 % of the initial power output could be detected. A closer voltammetric inspection of the individual electrodes (Supplementary Figures 15B and C) revealed that the turnover currents at the Py_2Ox /HRP based biocathode even after the addition of further glucose to the solution significantly decreased and are close to the background current. While the currents of the viologen units remain constant before and after the long-term test. The H_2 oxidation currents decreased by about 30 % indicating that parts of the hydrogenase were disintegrated at the bioanode. The results clearly demonstrate that the biocathode is the limiting electrode with respect to durability and that the bioanode shows the envisaged high stability even under operational conditions.

Besides, the authors claim that the bioanode and biocathode take severe damage under one chamber condition and do not clearly clarify which factors have influence on bioanode or biocathode.

The bioanode is stable in the one-compartment cell as shown by the experiments in Supplementary Figure 14: CVs under turnover and non-turnover conditions show identical current response before and after biofuel cell evaluation. However, the Py_2Ox /HRP biocathode suffers from severe damage during the test. We attribute this to the lower stability of the Py_2Ox system most likely caused by the local high concentration of H_2O_2 .

For the bioanode the oxidase/catalase system does not only remove O₂ but also protects the hydrogenase/polymer film from H₂O₂ damage (see also ref. 53). Experiments with inactive catalase showed a severely decreased stability of the bioanode. A full evaluation of a biofuel cell comprising an anode containing inactive catalase was not possible due to the fast disintegration of the bioanode (Supplementary Figure 16A). Cyclic voltammograms measured before and after the BFC test show that not only the activity for H₂ oxidation was lost but also that the redox polymer is destroyed by excess H₂O₂ (see new Supplementary Figure 17).

We added related discussion on this issue to the main text:

In contrast, when an inactivated catalase was used for the preparation of the double layer bioanode ([NiFeSe] based system), the evaluation of the fully assembled fuel cell was not possible due to the fast damage of the bioanode with the current dropping to zero within a short time (Supplementary Figure 16A). Comparison of cyclic voltammograms recorded before and after the biofuel cell test, unambiguously show that not only the hydrogenase itself but also the redox polymer matrix was destroyed due to the local high H₂O₂ concentrations during testing. This is evident from the complete loss in H₂ oxidation currents as well as the redox transition of the polymer-bound viologen moieties after the BFC test (Supplementary Figure 17A and B). Moreover, also the biocathode took severe damage and its catalytic response was significantly reduced after the BFC evaluation (Supplementary Figure 17C and D). This deactivation may be related to a voltage drop at the cathode caused by a change of the OCV due to the continuous disintegration of the bioanode leading to the destruction of the biocatalyst by applying extreme potentials. However, the exact nature of this deactivation of the biocathode remains unclear. Most importantly, the developed bioanode with full protection remains active even under harsh conditions (see e.g. Supplementary Figure 14A).

Finally, the superiority of bioanode in the protection of oxygen damage is not highlighted in the BFC system.

The proposed bioanode configuration leads to stable electrodes that can be used in two- and one-compartment configuration. Importantly, as mentioned several times in the manuscript, the experiments were performed with the bioanode as the current limiting electrode. Voltammograms of the bioanode recorded before and after the biofuel cell test unambiguously demonstrate that the activity of the hydrogenase was not significantly affected during testing. See Supplementary Figures and 11A and C as well as 14A. Optimization of the bioanode based on a fully protected hydrogenase based bioanode was the main goal of the study.

In the revised manuscript, my suggestions and questions are not well considered and answered. Thus I think the revised manuscript is not suitable to publish in Nat. Commun.

We hope that based on this additional revision and the performed experiments we could satisfy the reviewer and we hope that the reviewer is now able to support the publication in Nature Comm. We thank the reviewer for his constructive and helpful comments which indeed substantially helped to improve the manuscript.

Reviewer #2 (Remarks to the Author):

Most of my previous questions have been answered. Therefore, i would accept its publication in Nature Communications after a minor revision. I still have one comment which should be further clarified and discussed. It is still not clear to me which of the bioanode and the biocathode is limiting the power of the fuel cell. Looking at the current density of the cathode, the peroxidase electrode might be limiting the power of the fuel cell when comparing normalized current density. This is the reason why i asked in the previous review of this manuscript to give the power density referenced to

the average surface area of both the cathode and the anode since the limiting electrode might be the peroxidase-based cathode. Could the authors clarified this issue ?

The power output of any fuel cell will always be limited by the electrode that produces the smaller absolute current and not current density. Evidently, the absolute current can be adjusted by the size of the electrode. In this case the hydrogenase based bioanode with a nominal diameter of 3 mm/0.07 cm² shows an **absolute current response** of $\approx 20 \mu\text{A}$ (NiFe) and $\approx 40 \mu\text{A}$ (NiFeSe), see e.g. Supplementary Figures 11 A and C. The corresponding oversized biocathode (1 cm²) shows an **absolute current** of $\gg 100 \mu\text{A}$ (Supplementary Figure 11B and D). Which is at least 5 or 2.5 times higher than the maximal current at the bioanode. Hence, we made sure that the bioanode is always the limiting electrode which can be assured for all reported experiments. For a detailed report on the influence of the limiting electrode in hydrogenase-based biofuel cells see Armstrong et al., Energy Environ. Sci., 2013, 6, 2166 and Lojou et al. Electrochemistry Communications 23 (2012) 25–28.

Consequently, the maximum power that can be delivered by the biofuel cell is limited by the current that can be provided by the bioanode and hence if one wants to normalize the current by the area of an electrode the area of the limiting electrode has to be chosen. The size of the biocathode will not affect the power output and thus an average surface area which includes the surface of the biocathode would lead to non-comparable values. To make this clearer, we modified the text accordingly:

Cyclic voltammograms measured with the individual half cells show that the *absolute oxidation currents* of the bioanodes (Supplementary Figures 11A and C) are indeed smaller than the *absolute reduction current* at the biocathode (Supplementary Figures 11B and D) which assures the bioanode being the limiting electrode in all experiments.

Also, provide a photograph or a more precise scheme of the fuel cell would help the reader to understand the concept of this fuel cell on a more technological level.

A standard gas-tight three electrode electrochemical cell was used for the electrochemical characterization of the individual half cells. For biofuel cell test, two electrochemical cells were connected via a Nafion membrane, see below:

Reviewer #3 (Remarks to the Author):

This work reports the construction of a layered hydrogenase-based bioanode, which aims to protect sensitive hydrogenases both from potential-induced inactivation and O₂ damage. A viologen-based polymer is used as a redox mediator matrix and as a buffer layer to protect hydrogenase from potential inactivation. A second layer made of glucose oxidase and catalase entrapped in a redox silent polymer serves to consume O₂. This strategy aims to overcome the issues rose by the use of the viologen-based polymer, i.e. requirement of a thick film, and waste of electrons for O₂ reduction. Stability of the bioanode over around 1 hour is demonstrated using electrochemistry on two

different hydrogenases. A new issue however comes from the formation of gluconic acid which decreases the electrolyte pH to such an extent that the hydrogenase is denatured. The use of another oxidase for glucose oxidation with production of a neutral ketone is proposed, which however does not allow to efficiently remove O₂. Nevertheless, the experiments are globally well conducted and will interest scientists working on enzyme bioelectrochemistry.

We are very happy about the in-depth evaluation by the reviewer and we highly appreciate his/her positive evaluation.

One main concern is the poor stability of the bioanode in the absence of the second protective layer compared to previous works by the same group (a few hours against one month in Nature Chem. 2014), which needs to be discussed in the present work.

Please note that the stability of the bioanode reported in the Nature Chem. paper from 2014 was tested under **oxygen-free** conditions. The here presented bioanode shows an excellent stability under anaerobic conditions as well: only a slight decrease of the H₂-oxidation current was observed after 18 h, see Figure 3A, black line. The stability of the here proposed system is comparable to the stability of the previously reported polymer/hydrogenase film under O₂ free conditions.

Of course, under aerobic conditions the stability is less, since the oxygen front is penetrating the entire polymer film with time and destroys the active layer over time, see Figure 3A, red line. However, such a long-term measurement was not performed in the previous publication published in Nature Chem, 2014 and a comparison is not possible. Moreover, the use of the oxidase/catalase protection system dramatically increases the lifetime of the bioanode as can be seen from Figure 3A and B (cf. red line with blue and green lines) clearly demonstrating that the proposed bienzymatic system shows the envisaged protection effect.

The so-built bioanode is coupled to a biocathode based on horseradish peroxidase which reduces H₂O₂ formed in situ by co-immobilized glucose oxidase. The biofuel cell delivers power densities in the range of 500 μW/cm² at 0.85 V. Although this power output is not very impressive, such a biofuel cell concept is new and may open new avenue if the issue linked to strong acidification of the electrolyte is solved in the future. The concept of the protective bilayer bioelectrode was recently developed for glucose oxidase, and the viologen-redox polymer was applied to various hydrogenases. But this is the first time such a construction is used to provide a full protection to hydrogenase. In that sense, this work is innovative, and worth being published, after the following points are addressed.

Points to be addressed before publication:

- A discussion is required about the spatial organization of the two layers. How to insure stable layered structure and no fusion of the two layers once rehydrated ?

We cannot fully exclude intermixing of the active hydrogenase/polymer and the oxidase/catalase/polymer layer upon rehydration. However, if the hydrogenase polymer layer would reach the film-electrolyte interface, a current decrease in analogy to the single layer system upon O₂ addition would be visible due to the fact that the enzymatically generated viologen radical cation would react with any oxygen reaching this layer and the electrons for the formation of the viologen radical cation would be lost. Since no current decrease is observed upon addition of oxygen, one can conclude that the inner viologen-based redox polymer/hydrogenase layer is fully covered by the top layer which is used in excess compared to the active layer preventing any oxygen to reach the active layer. Moreover, if the oxidase and the catalase would come in contact with the viologen-based polymer layer no interaction possibility would arise. The catalase would still disproportionate the H₂O₂ generated by the oxidase which cannot be wired via the viologen due to the too low redox potential. We modified the main text accordingly, to make this point more clear.

Cyclic voltammograms in phosphate buffer (0.1 M, pH 7.4) containing glucose (50 mM) under argon (Supplementary Figure 1B, black dashed line) and under air (red line) show the unchanged reversible signal of the polymer-bound viologen moiety. A significant O₂ reduction via the low potential viologen based mediator¹⁵, as it was observed for voltammograms recorded without glucose in solution (blue line), was absent indicating that O₂ is fully removed in the top layer. This also indicates that both layers do not intermix significantly during rehydration since no direct exposure of the viologen-modified polymer/hydrogenase layer to the bulk electrolyte is detected. In contrast to the experiments in quiescent solutions, slightly enhanced cathodic currents were observed due to enhanced mass transport, when a mixture of 5 % O₂ and 95 % Ar is purged through the electrolyte. Trace amounts of O₂ seem to reach the underlying active layer under these conditions (Supplementary Figure 1C).

- As discussed in a previous paper, one important point in order to understand and control the protective effect of the biozymatic layer is the thickness of the different layers. Information and discussion about this parameter is required.

At a first glance, the polymer layer thickness seems to be an important characteristic. However, even more important than the absolute thickness are the diffusion properties of the substrate and electron transfer properties within the film etc., which will finally all determine the behavior of the current response of the electrode. According to the classification reported by Bartlett (ref. 50) and by us (ref. 18) the films can be classified as regime III/case III for which the current is mass transport limited and depends on the H₂ concentration. This is evidenced by the calibration graph depicted in Supplementary Figures 2C and D. Hence, if oxygen protection would be based solely on H₂-oxidation, a decrease in the steady state currents in chronoamperometric experiments would be visible upon O₂ addition. Since this is not the case, we conclude that the proposed O₂ removal system shows the envisaged full protection of the hydrogenase based bioanode. To make this even more clear we changed the text accordingly:

This behavior corresponds to regime III or case III following the notation described in refs. ¹⁸ and ⁵⁰, respectively. Note that for this behavior a current decrease is expected upon O₂ addition if no additional protection system is used^{15,18}.

- Different hydrogenase batches have been used which could explain some discrepancies in the results. Gels showing the purity of these different batches as well as specific activity are then required.

All enzymes that were used in this work showed the characteristic features reported for those biocatalysts in refs 52 and 53 (NiFe) and 48 (NiFeSe). A detailed description of the purification process of the enzymes can be found in the cited literature. The activities of the enzymes have been added to the experimental section. Moreover, we want to emphasize that the behavior of the individual electrodes is independent of the absolute current. Thus, an at least qualitative comparison of different electrodes measured under different conditions is still possible.

The [NiFe] hydrogenase from *Desulfovibrio vulgaris* Miyazaki F (DvMF-[NiFe]) was isolated and purified according to ref.^{52,53}. It was stored in MES buffer at pH 6.8 at -20 °C with a concentration of 200 µM. The H₂ production activity of this type of hydrogenase was determined to be 610 U mg⁻¹.⁵³ However, the activities of the individual batches vary and electrodes show different activities towards the oxidation of H₂ (±50 %, with respect to the average value). The recombinant form of the [NiFeSe] hydrogenase from *Desulfovibrio vulgaris* Hildenborough (DvH-[NiFeSe]) was isolated and purified as described previously in ref.⁴⁸ The activity for H₂ formation was estimated to be 4000 – 5700 U mg⁻¹, depending on the batch with variations of up to 320 U mg⁻¹ for each single batch. Hence, the activity of modified electrodes varies by ±90 % (with respect to the average value). The

enzyme was stored at $-80\text{ }^{\circ}\text{C}$ in 20 mM Tris-HCl at pH 7.6 with a concentration of $14\text{-}15\text{ }\mu\text{g }\mu\text{L}^{-1}$ (159-170 μM). Note that rather high concentrations of the enzymes were used to achieve high biocatalyst loading on the electrode surface while keeping the volumes used during the drop cast process rather small to minimize drying time and to facilitate the modification of the 3 mm diameter electrodes.

Although, the activity of the individual electrodes towards H_2 oxidation varies for different enzyme batches which leads to a variation in the absolute current response, all electrodes show the same trend under different conditions thus allowing an at least qualitative comparison of the individual systems.

Reviewer #4 (Remarks to the Author):

In this work by Ruff et al., the construction of novel multilayer redox-polymer/biocatalyst electrode surfaces are described with perceived application in fuel cells, in which hydrogen is reduced at the anode by a biocatalyst, hydrogenase, which is well known to be deactivated by oxygen. The experiments are very carefully performed and described in very good detail, supporting the conclusions by the authors. The main advance of this work, compared to the previous state of the art, lies in the fact that the hydrogenase-modified anode is protected from oxygen damage. The authors have previously described protection mechanisms against oxygen damage that relied on relatively thick polymer layers on the anode. In this work, however, a different strategy is employed that relies on an oxygen reducing protective enzyme layer, which deposited on top of the hydrogenase layer. The main advance of the new strategy – to my opinion - is that thinner hydrogenase-polymer layers can be used potentially reducing the amount of hydrogenase catalyst.

We are very happy about the in-depth evaluation by the reviewer and we highly appreciate his/her positive evaluation.

There are couple of main issues with the strategy employed that need to be addressed prior to publication.

1) The authors describe their fuel cell as a “hydrogen/hydrogen peroxide fuel cell”. However, their fuel cells is NOT a hydrogen/hydrogen peroxide fuel cell, but a hydrogen/glucose fuel cell. The cell as described by the authors would not be able to be fueled directly by hydrogen peroxide as the catalase on the anode would consume the hydrogen peroxide in a loss making side reaction.

This is indeed a very important point, however, we disagree here with the reviewer. If H_2O_2 was added instead of glucose, which is leading to enzymatically generated H_2O_2 in both electrodes, the biofuel cell would still function due to the fact that H_2O_2 is the electron acceptor at the cathode side. Evidently, on the anode side the outer-layer O_2 removal system would not function, but the catalase in the outer layer would still protect the hydrogenase from damage by H_2O_2 . Evidently, it is by far more elegant to locally produce the required H_2O_2 in close proximity of the biocathode using enzymatic glucose oxidation and to simultaneously use the glucose in the outer O_2 removal layer. There are two possible definitions for denoting the corresponding fuel cells, i.e. (i) what is added (glucose) vs. what does actually consume (or provide) electrons (H_2O_2). We were following the second definition, but we agree that this may be in the specific case too simple and we modified the title and also the main text to follow the reviewer’s suggestion. Throughout the whole manuscript we use now the proposed “ H_2 /glucose(H_2O_2) biofuel cell” terminology:

A fully protected hydrogenase/polymer based bioanode operating in a high performance H_2 /glucose(H_2O_2) biofuel cell

Furthermore, the authors have not shown that the bioanode can withstand hydrogen peroxide at, for instance, 2 mM concentration. The fact that this is hydrogen/glucose fuel cells needs to be made clear in the title, abstract and main manuscript.

As it was already mentioned in the text, the oxidase/catalase system does not only remove oxygen but also protects the active layer from H_2O_2 , which is also formed when the viologen mediator is reducing O_2 in a conventional single layer configuration. However, high concentrations of H_2O_2 lead to a deactivation of the hydrogenase (see also ref. 53) and also destroy the redox polymer. This was further evidenced by experiments for which a deactivated catalase was employed (see answers to Reviewer 1) for which a strong current drop was observed during the fuel cell evaluation (Supplementary Figure 16 and 17).

CVs recorded with the protected bioanode after the biofuel cell tests in a one compartment configuration (Supplementary Figure 14) show that the activity of the hydrogenase and also the electrochemical response of the viologen modified polymer are not altered. Thus, we conclude that the even under H_2O_2 production (within the protection layer at the anode and at the biocathode) the proposed protection system is able to fully remove O_2 and H_2O_2 . The title/abstract/main text has been changed accordingly, see 1).

2) Previous enzyme-based fuel cells described in literature are often hydrogen/oxygen fuel cells. In hydrogen/oxygen fuel cell it is perceivable that oxygen leaks to the anode and hence oxygen can inactivate the hydrogenase catalyst, and thus oxygen damage needs to be protected against. However, in the work by Ruff et al., a hydrogen/glucose fuel cell has been constructed. What is the author's argument that the anode needs to be protected against oxygen in a hydrogen/glucose fuel cell?

In the proposed biofuel cell O_2 is indeed as well necessary, since it is required for the in-situ generation of H_2O_2 from glucose and O_2 by glucose oxidase or pyranose oxidase that is co-immobilized at the cathode together with horseradish peroxidase. Without O_2 , no H_2O_2 would be generated even in the presence of glucose. Hence, O_2 is essential for the functioning of the biocathode and thus it can be expected that at least trace amount of O_2 will reach the bioanode. To make this even more obvious we modified the text accordingly.

Introduction:

Here, we report the design of a fully protected polymer multilayer based hydrogenase bioanode combined with an oxidase/HRP biocathode for the fabrication of a H_2 powered biofuel cell that consumes H_2O_2 as the oxidant which is generated *in-situ* from O_2 by an oxidase and in the presence of glucose to keep the concentration of harmful H_2O_2 low. The device shows an extraordinary high OCV and remarkable current densities.

Results:

The oxidase/catalase O_2 removal system is fueled by glucose which simultaneously acts as the reactant for the *in-situ* generation of the oxidant H_2O_2 at the oxidase/HRP based biocathode. H_2O_2 will only be formed by the oxidase in the presence of O_2 . Consequently, also for the proposed H_2 /glucose fuel cell, protection of the bioanode from O_2 is indispensably required.

3) The protection layer on the anode consumes glucose, but glucose is also its fuel at the cathode. How much glucose is consumed at the anode has not been measured by the authors. An important aspect of a fuel cell is the Faradaic yield and this needs to be reported by the authors, especially seen the aspect just described.

Glucose is added in excess which is required for keeping the anode protected and the cathodic currents high. With 5 mM glucose in solution no current decrease was observed during continuous operation of 6 h (see Supplementary Figure 10B). Note that this time span is considerably longer than all typically shown biofuel cell tests.

Moreover, we want to emphasize that in a potential technical application, one has to assure that the glucose concentration does not fall below a threshold which would either make the cathode to be the limiting electrode (or even stop the reaction at the electrode in case all glucose has been consumed) or compromise the protection of the hydrogenase anode.

We agree that the Faradaic yield is an important characteristic for conventional fuel cells. However, in the proposed system the exact concentration of the gases H_2 (fuel for the bioanode) and O_2 (for the in-situ generation of H_2O_2 by the concomitant oxidation of glucose by the corresponding oxidase) which were just bubbled through the electrolyte (exact pressure and amount are not known) is not known. Thus, a reliable value for the faradaic yield cannot be provided. However, the bioanodes even in the two-layer configuration show current densities that are similar to our previously reported systems (see refs. 15-17). Also, the biocathode show comparable current responses to previously reported systems (refs. 37 and 38).

We added a note to the main text:

The current responses of the bioanodes are similar to previously reported polymer/hydrogenase based bioanodes.^{15,17}

And

Obviously, both enzymes are able to produce substantial amounts of the oxidant H_2O_2 as indicated by the pronounced catalytic waves under turnover conditions (red lines, note that the current response of the carbon cloth based electrodes is similar to our previously described systems based on CNT/CMF modified carbon rods^{37,38}).

Minor issues:

a) In Line 150, the authors write that “a large excess of the Gox/CAT/P(SS-GMA-BA)” layer was used in the preparation of the data shown in Supplementary Data Figure 1. Can the authors clarify if this large excess was used in all experiments or only for Supplementary Data Figure 1.

Yes, the redox silent polymer P(SS-GMA-BA) was always used in excess (in terms of mass) compared to the viologen modified polymer P(N_3 MA-BA-GMA)-vio. We added a note on this to the text:

For this, pristine P(N_3 MA-BA-GMA)-vio films were coated with a large excess of the GOx/CAT/P(SS-GMA-BA) mixture in a drop cast process to form a P(N_3 MA-BA-GMA)-vio//P(SS-GMA-BA)/GOx/CAT double layer system (note that for all experiments a large excess of the second layer with respect to polymer mass was used to ensure the full coverage of the active hydrogenase layer, for composition and polymer/enzyme ratios see Methods part).

b) In Line 200-202, the authors note that the “for lower hydrogen contents the difference seems to be lower” and ascribe this to “change in film swelling”. However, from Figure Supplementary Data 2C and 2D, it is clear that the current is mass transport limited. I might have misunderstood, but is the smaller difference between Figure 2C and 2D not simply due to the fact that both currents are limited by mass transport of hydrogen and thus very similar in both cases, independent on the biocatalyst used?

We thank the reviewer for this important comment. Since the currents are mass transport limited at this H_2 level, both currents can indeed be similar. We changed the text accordingly:

However, for lower H_2 contents the difference seems to be lower (cf. Figure 2C and 2D) which may be related to the fact that the currents are limited by the mass transport at these low H_2 levels.

c) In line 234, the authors suggest the “protons generated by the conversion of hydrogen” could contribute to the lowering of the electrolyte pH. However, in this case a three electrode cell has been

used with a Pt counter electrode (line 444), thus for every hydrogen oxidised at the working electrode, either (half a) oxygen is reduced at Pt electrode or two protons reduced to hydrogen. Would the authors agree that the Pt counter electrode is likely to prevent pH changes due to hydrogen oxidation by hydrogenase?

Indeed, the used wording is somehow misleading. Of course, the global pH value of the bulk solution will not change. However, the local pH value within the film may drop to a critical value at which a chemical disintegration of the enzyme takes place. We changed the text accordingly:

In combination with the protons generated by the conversion of H₂ at the bioanode, the local concentration of H⁺ within the polymer/enzyme film exceeds the buffer capacity in the solvated hydrogel and the pH drops to a critical value at which the enzyme is disintegrated.

d) In Line 265-267, the authors claim an “almost identical intensity indicating that that the polymer matrix is not affected under these conditions”. However, in Supplementary Figure 7B, about 20-30% of the viologen signal seems to have disappeared while the hydrogen oxidation current is reduced by ~60%. As there is no easy way to correlate the “density” of viologen in the polymer to the electronic coupling of the hydrogenase, it cannot be excluded that damage the viologen in part contributes to the reducing in current.

The reviewer is right, we cannot exclude that also viologen degradation may lead to a reduced current output. We modified the text accordingly:

Cyclic voltammograms recorded under argon after the chronoamperometric experiment show slightly decreased signals of the viologen units indicating that also the polymer matrix, although to a lower extent when compared to the GOx based system, is affected under these conditions (Supplementary Figure 7). Hence, not only enzyme deactivation but also polymer degradation may contribute to the observed current decrease.

e) In Line 370-272, can the authors acknowledge that the drawback of Py₂Ox is that is less able to prevent oxygen damage and hence only protects up to 3% oxygen.

We changed the text following the reviewer’s suggestion.

However, the higher stability is counterbalanced by a lower protection efficiency of the Py₂Ox system (3 % O₂) compared to the GOx based system (5 % O₂).

f) Line 444, could the authors give a reference in which the standard protocol for polishing is described. It is know that details of polishing can have large effect on electrochemical behaviour.

We added the procedure to the experimental part. The electrodes were subsequently polished with several alumina/water suspensions of decreasing particle grain size prior to the experiments.

Prior to each experiment, the glassy carbon working electrodes were subsequently polished using several alumina/water suspensions with decrease particle grain size (going from 1 μm, via 0.3 μm to 0.05 μm), each with ≈1 min. After each polishing step the electrodes were thoroughly rinsed with water. After final polishing step the electrodes were again washed with water and dried in an argon stream to remove any dust particles.

REVIEWERS' COMMENTS:

Reviewer #1 (Remarks to the Author):

Comments to the author

In this revised manuscript, the H₂/ glucose (H₂O₂) enzymatic biofuel cell is well described and my suggestions and questions are well considered and answered. Besides, the comments from other 3 reviewers are well organized. Thus I think the revised manuscript is a good work on hydrogenase catalyst research and suitable to publish in Nature communications.

Reviewer #2 (Remarks to the Author):

I am ok with all the responses made by the referees. Of course, on the matter of absolute currents, this is ok because both electrode are closed in terms of size. On a technological point of view, this discussion on absolute current and limiting electrode would not work if one electrode would be much bigger than the other. In my opinion, this original work is now appropriate for publication.

Reviewer #3 (Remarks to the Author):

In this revised version, the authors have addressed most of my issues. One point is still confusing for me. It concerns the comparative stability of the current bioanode compared to that one reported in Nat. Chem (2014). If I well understood the experiments conducted 4 years ago, the stability of the bioanode was tested during 15 days in the membrane less fuel cell fed with 95%H₂/5%O₂. The bioanode was quite stable over that period, which contrasts with the poor stability reported in the current work. Could the authors provide one clear explanation, their work could be published.

Reviewer #4 (Remarks to the Author):

The authors have carefully considered and addressed all my major comments (and minor comments). I fully support publication of this work, as is, in Nat. Comm. As described in my previous comments to the authors, this works shows an elegant novel construction of a bioanode (and biocathode). The use of multi-layers to counter specific drawbacks of certain biocatalyst, while exploiting their specific advantages, will certainly be of interest to everybody in the fuel cell and bioelectrochemistry communities.

Response to Reviewers (in blue):

We thank Reviewers 1, 2 and 4 and we are happy that they find now our manuscript suitable to be published in Nature Communications.

Reviewer #1 (Remarks to the Author):

Comments to the author

In this revised manuscript, the H₂/ glucose (H₂O₂) enzymatic biofuel cell is well described and my suggestions and questions are well considered and answered. Besides, the comments from other 3 reviewers are well organized. Thus I think the revised manuscript is a good work on hydrogenase catalyst research and suitable to publish in Nature communications.

Reviewer #2 (Remarks to the Author):

I am ok with all the responses made by the referees. Of course, on the matter of absolute currents, this is ok because both electrode are closed in terms of size. On a technological point of view, this discussion on absolute current and limiting electrode would not work if one electrode would be much bigger than the other. In my opinion, this original work is now appropriate for publication.

Reviewer #3 (Remarks to the Author):

In this revised version, the authors have addressed most of my issues. One point is still confusing for me. It concerns the comparative stability of the current bioanode compared to that one reported in Nat. Chem (2014). If I well understood the experiments conducted 4 years ago, the stability of the bioanode was tested during 15 days in the membrane less fuel cell fed with 95% H₂/5% O₂. The bioanode was quite stable over that period, which contrasts with the poor stability reported in the current work. Could the authors provide one clear explanation, their work could be published.

The stability of the bioanode analyzed 4 years ago was **NOT** tested in a membrane less fuel cell assembly but in a single bioanode compartment and without O₂ content in the gas feed, which was composed only of H₂.

In this work, the stability of the fuel cell was tested with O₂ in the gas feed. Thus, a direct comparison is not possible for two reasons: (I) comparison between single half cell (bioanode in Nature chem from 2014) and biofuel cell (this work) and (II) the gas feed in the previous contribution was H₂ only and in this work a combination of H₂ and O₂ was used.

However, as already mentioned in the text, the operational stability of the fuel cell is similar to a fuel cell that is using a thermostable and O₂ tolerant hydrogenase working in a DET regime (see ref. 52). We want to emphasize again that in this work a highly **O₂ sensitive hydrogenase** was used. To make this even more clear we modified the discussion accordingly:

The operational stability is similar to a high current density hydrogen biofuel cell that is working in a DET regime using a thermostable and O₂ tolerant hydrogenase.⁵²

Reviewer #4 (Remarks to the Author):

The authors have carefully considered and addressed all my major comments (and minor comments). I fully support publication of this work, as is, in Nat. Comm. As described in my previous comments to the authors, this work shows an elegant novel construction of a bioanode (and biocathode). The use of multi-layers to counter specific drawbacks of certain biocatalyst, while exploiting their specific advantages, will certainly be of interest to everybody in the fuel cell and bioelectrochemistry communities.